# International Music Therapists’ Perceptions and Experiences in Telehealth Music Therapy Provision

**DOI:** 10.3390/ijerph20085580

**Published:** 2023-04-19

**Authors:** Amy Clements-Cortés, Marija Pranjić, David Knott, Melissa Mercadal-Brotons, Allison Fuller, Lisa Kelly, Indra Selvarajah, Rebecca Vaudreuil

**Affiliations:** 1Faculty of Music, The University of Toronto, Toronto, ON M5S 1A1, Canada; 2Holland Bloorview Kids Rehabilitation Hospital, Toronto, ON M4G 1R8, Canada; 3Seattle Children’s Hospital, Seattle, WA 98105, USA; 4Escola Superior de Música de Catalunya, 08013 Barcelona, Spain; 5School of Humanities and Communication Arts, Western Sydney University, Penrith, NSW 2751, Australia; 6Irish World Academy of Music & Dance, University of Limerick, V94 T9PX Limerick, Ireland; 7Department of Music, Universiti Putra Malaysia, Malaysian Society for Music in Medicine, Serdang 43400, Malaysia; 8Independent Researcher, Worcester, MA 01604, USA

**Keywords:** telehealth, music therapy, telehealth music therapy (TMT), COVID-19, survey

## Abstract

The use of telehealth within music therapy practice has increased through necessity in recent years. To contribute to the evolving evidence base, this current study on Telehealth Music Therapy (TMT) was undertaken to investigate the telehealth provision experiences of music therapists internationally. Participants completed an anonymous online cross-sectional survey covering demographics, clinical practice, telehealth provision, and telehealth perceptions. Descriptive and inferential statistics, in combination with thematic analysis, were used to analyze the data. A total of 572 music therapists from 29 countries experienced in providing TMT took part in this study. The results showed that the overall number of clinical hours (TMT and in-person hours combined) declined due to the pandemic. Participants also reported reduced perceived success rates in utilizing both live and pre-recorded music in TMT sessions when compared to in-person sessions. Although many music therapists rose to the challenges posed by the pandemic by incorporating TMT delivery modes, there was no clear agreement on whether TMT has more benefits than drawbacks; however, reported benefits included increased client access and caregiver involvement. Furthermore, a correlation analysis revealed moderate-to-strong positive associations between respondents who perceived TMT to have more benefits than drawbacks, proficiency at administering assessments over telehealth, and perceived likelihood of using telehealth in the future. Regarding the influence of primary theoretical orientation and work setting, respondents who selected music psychotherapy as a primary theoretical orientation had more experience providing TMT prior to the pandemic while those primarily working in private practice were most inclined to continue TMT services post-pandemic. Benefits and drawbacks are discussed and future recommendations for TMT are provided.

## 1. Introduction

Since the inception of remote healthcare provision in the mid 1900s and its evolution to date, there have been a range of different terms that describe these services, including telemedicine, telerehabilitation, teleintervention, and telehealth [1]. Telemedicine is the delivery of medical care and the provision of general health services from a distance [2] Telerehabilitation is a branch of telemedicine that provides patient care following an acute phase of disease with a focus on reducing hospital stay length and re-hospitalization and increasing patients’ ability to perform rehabilitative tasks at home [3]. Teleintervention is associated with the provision of early intervention services via teleconferencing technology, which has been primarily used to increase access to services for families of children with special needs, especially those who are deaf or hard of hearing [4] Telehealth is the “delivery and facilitation of health and health-related services including medical care, provider and patient education, health information services, and self-care via telecommunications and digital communication technologies” [5].

Many of these terms have often been used interchangeably; however, as the area of remote healthcare continues to evolve, telehealth is becoming a commonly used and recognized terminology that refers to a broader scope of remote healthcare activities and services [1]. Additionally, the survey results presented in this article indicate that music therapy professionals from across the world use the term telehealth when referring to the delivery of remote music therapy services. Therefore, this article will refer to the remote delivery of healthcare services as telehealth and the remote delivery of music therapy as telehealth music therapy (TMT). 

In recent decades, telehealth has emerged as an efficient and effective way for healthcare providers and patients to communicate and exchange important information through secure messaging, tele- or video conferencing, and, additionally, to assess, treat, and monitor health issues, and prescribe medication [6]. Recent years have necessitated an increased demand for the remote delivery of healthcare services on a global scale. Specifically, telehealth has reduced the risk of infection during the COVID-19 global pandemic by limiting physical contact and better accommodating individuals with immune deficiency and other factors that complicate in-person treatment and travel to and from appointments. Additionally, telehealth has generated increased access to healthcare services and treatment opportunities in resource-poor countries during the pandemic, especially those with a scarcity of specialists in underserved regions. 

As telehealth has been more commonly accepted and implemented in healthcare fields and across systems since the turn of the 21st century, it has also gained visibility in the creative arts therapies, including music therapy [7,8]. Music therapy practices have been successfully adapted to telehealth and yielded positive participant responses, including reports of decreased pain, anxiety, and depression [9]. 

### Literature Review

Prior to the COVID-19 pandemic, TMT was limited and just beginning to be explored. Articles described TMT being provided to active duty military and veteran populations [7,8,9,10], parents of hospitalized infants in neonatal intensive care units [11], adolescents with Asperger’s Syndrome who lack access to in-person services due to residing in remote or rural communities [12,13], and children with hearing loss [14]. Global public health measures initiated at the onset of the pandemic limited the in-person provision of music therapy services, which led to rapid innovations in TMT [15]. 

A policy statement in support of telehealth from the American Music Therapy Association (AMTA) stated that “Telehealth/therapy is subject to the same AMTA Standards of Clinical Practice established for face-to-face music therapy” [16]. Increased understanding of the clinical practice components and competencies required to facilitate TMT has been disseminated through scoping reviews, case studies, professional perspectives, surveys, webinars, and guidance from various music therapy organizations around the globe [17,18,19]. A survey of music therapists in the United States (U.S.) by Gaddy et al. [20] initially reported the prevalence of music therapists’ shift to providing remote services and described the impact of COVID-19 on employment, service delivery, and measures of stress and hope. A majority of respondents (54.81%) reported transitioning services to TMT delivery, with 87.49% reporting that they felt hopeful about the profession of music therapy, with the potential of telehealth services noted as one of the top three comments. 

In December of 2020, the AMTA shared highlights from a survey of music therapists [21]. Nearly two-thirds of all respondents reported transitioning their practices to telehealth and 74% indicated that their clients/consumers had a positive response to TMT, with some reporting that “certain clients/consumers are thriving in ways that did not happen when seen in person” [21]. Additional reported benefits included greater involvement of family members in sessions and increased support felt by co-workers employed in residential treatment facilities [21]. 

Cole et al. [22] surveyed music therapists certified in neurologic music therapy (NMT) and found that telehealth treatment successfully transferred across all domains (i.e., cognitive, sensorimotor, speech/language, psychosocial); however, fewer therapists reported implementing sessions within the sensorimotor domain. Additionally, positive outcomes were correlated with increased caregiver involvement [22]. An international survey study by Agres et al. [23] explored technological questions related to adaptation to TMT service delivery and found significant geographical differences, with music therapists in North America reporting greater success in transitioning their services. This difference is potentially due to increases in demand for services, the guidance and support provided in TMT transition, and client access to the required resources for TMT such as digital devices (e.g., smartphones, tablets, computers), internet connectivity, and musical instruments. Furthermore, there were statistically significant differences in favorable responses from music therapists practicing in North America compared with respondents from Europe or Asia/Oceania [23]. 

Baker and Tamplin [24] surveyed 60 registered music therapists (RMT) and 16 clients/consumers in Australia. Benefits reported by RMTs include increased access for clients living in remote areas, family involvement in sessions, reduced cancellations and damage to instruments, and increased communication and eye contact from some clients. Clients/caregivers reported a perceived sense of safety from reduced risk of infection and found TMT to be either “somewhat” or “very effective” (81.3%) [24]. Reported challenges for RMT’s and clients/caregivers included client engagement (difficulty observing non-verbal cues), technical limitations (device availability, internet access, sound quality), and need for increased caregiver support. Despite the noted challenges, 78% of RMT’s and 43.8% of clients/caregivers reported they would continue using TMT after Covid-19 restrictions were lifted, with both groups citing increased access [24].

Wilhelm and Wilhelm [25] (2022) conducted a survey of music therapists providing TMT to older adults, which yielded similar results regarding the reported benefits of access, ability to see clients/consumers, and inclusion of family members in the session, as well as technical challenges that limited opportunities for collective music-making, lack of privacy for clients/consumers, and unfamiliarity with the technology. 

A scoping review provided a summary of TMT adaptations, benefits, and challenges across 10 articles [26]. Adaptations included therapeutic goals, methods and techniques, and the therapeutic positioning of the client and set-up of the music therapist’s clinical space to deliver the session. Benefits included increased client access to services, including providing those who are not independently ambulatory having with the opportunity to access group music therapy from their respective locations, and the ability to involve family members/caregivers in clinical sessions [26]. Challenges included an inability to synchronize musical interactions, poor sound quality, and the lack of ability to see and touch the instruments [26]. 

Knott and Block [27] identified that while there are benefits to TMT, individual suitability must be considered by the music therapist. For example, some older adults with cognitive, communication, or technical limitations have been identified as having telehealth unreadiness [28]. Similarly, music therapists providing telehealth have found that, depending on level of disease progression and disposition, some clients/consumers with dementia may not be appropriate for TMT [29,30]. 

Key themes of TMT have emerged, including the benefits of increased access to services and engagement of caregiver networks, and limitations related to the technical and relational challenges of the telehealth environment. Promising new work suggests that leveraging the possibilities inherent in the functionality of telehealth interaction may lead to improved therapeutic relationships and outcomes [31,32]). Additionally, the representation of student [33] and educators’ [34] perspectives suggests that the increased inclusion of telehealth content in the future academic preparation of music therapists is warranted. 

## 2. Method

### 2.1. Purpose of Study

This descriptive survey study [35], utilizing an anonymous survey via Google Forms, sought to assess music therapists’ perceptions and experiences in telehealth music therapy provision with a mixture of qualitative and quantitative data. (See Appendix A: Complete Survey). 

### 2.2. Research Questions

What are the perceptions and clinical practice experiences of music therapy professionals from around the globe in providing Telehealth Music Therapy (TMT) services?What are the benefits and challenges of TMT, and what is needed to advance this practice?Does a music therapists’ primary theoretical orientation and work setting contribute to perceptions and trends associated with the successful implementation of TMT services?

### 2.3. Participants

Informed consent was obtained on the first page of the survey before participants accessed the survey questions. Inclusion criteria included that the music therapist respondents: were certified music therapists or equivalent (must be permitted to practice music therapy in their region); were able to read and understand English; and had administered music therapy via telehealth for a minimum of 10 individual and/or group sessions. Participants voluntarily completed the survey, and no incentive was given. Demographic information was collected without personally identifiable information. Participants had the opportunity to provide their email address at the end of the survey if they wished to participate in a follow-up interview for a subsequent research study. If interested, the participants were prompted to enter their contact information on the following page after completing the survey. This study received ethical approval from the University of Toronto Research Ethics Board. 

### 2.4. Recruitment

Music therapy associations in the eight global regions of the World Federation of Music Therapy (WFMT) were asked to send the invitation to their members via social media posts on their Facebook, Twitter, and Instagram platforms. The research team also shared the invitation with their networks, Facebook, LinkedIn, Twitter, and Instagram music therapy groups (*n* = 70) and purchased an email list (*n* = 969) from the Certification Board for Music Therapists (CBMT) to invite board certified music therapists to complete the survey. The period of data collection was from 13 January to 30 July 2022. 

### 2.5. Data

This cross-sectional survey consisted of 33 closed and open-ended questions divided into four sections: (1) Demographics; (2) Clinical Practice; (3) Telehealth Provision; and (4) Telehealth Perceptions. The survey was administered by Google Forms and took approximately 10–15 min to complete (see Appendix A). 

### 2.6. Data Analysis

Both qualitative and quantitative data were collected and analyzed. The number of participants/response counts varied across questions, as some of the questions were not mandatory. Descriptive statistics were used to summarize the characteristics of this data set, and inferential statistics were used to further examine the relationships between variables. 

Data from closed-ended questions were coded and then analyzed using the SPSS (Statistical Package for Social Sciences) software version 28. Descriptive statistics were reported using means and standard deviations for continuous variables and frequency with percentages for categorical variables. The chi-square test, the Wilcoxon signed-rank test, and correlation analysis were utilized where applicable to identify the trends and factors that contributed to the successful implementation of telehealth music therapy services. All tests were two-tailed, with a significance level of alpha < 0.05.

#### 2.6.1. The Wilcoxon Signed-Rank Test

The Wilcoxon matched-pairs signed rank test was used to analyze the median of differences between (1) the number of clinical hours per week before and since the COVID-19 pandemic, (2) the perceived success of using live music during in-person music therapy sessions compared to TMT, and (3) the perceived success of using pre-recorded music during in-person music therapy sessions compared to TMT.

#### 2.6.2. Chi-Square Test of Independence

The chi-square test of independence was performed to determine whether there was a significant association between categorical variables to further our understanding of the experiences and trends that have emerged in music therapy practice since COVID-19. First, a chi-square test was used to assess the relationship between the primary theoretical orientation utilized when providing TMT (the top four being humanistic, eclectic and/or integrative, neurologic music therapy, music psychotherapy) and (a) global region of respondents, (b) education level, (c) years of practice, (d) experience with TMT prior to the pandemic, (e) changes in client’s goal areas due to the transition to TMT, (f) perceptions of caregiver involvement, and (g) perceptions of providing TMT services post-pandemic. Additionally, the chi-square test of independence was performed to examine the association between the primary work setting (e.g., private practice, healthcare/medical facility, academia) and selected variables. Ordinal responses were grouped into three categories: (1) strongly disagree/Disagree, (2) neither agree nor disagree, and (3) strongly agree/agree; or (1) not at all/for some clients/consumers, (2) unsure, and (3) likely/yes, definitely. When more than 20% of the expected counts were less than five, Fisher’s Monte Carlo Method [36] was reported with a 99% confidence interval. Fisher’s exact test was computationally infeasible due to the large sample size (*n* = 572). Significant effects were further determined with post hoc tests based on adjusted standardized residuals (z-values). Estimated *p*-values were adjusted using the Bonferroni correction method [37] to control for type I error inflation.

#### 2.6.3. Correlation Analysis

Spearman’s rank-order correlations were computed to further examine the relationships between ordinal variables [38]. A total of seven variables were included: (1) number of years practicing as a music therapist, (2) telehealth perceptions (i.e., “Telehealth music therapy has more benefits than drawbacks”), (3) education level, (4) change in client’s goal areas (i.e., TMT vs. in-person), (5) ability to administer assessments over telehealth, (6) future telehealth (i.e., “Do you anticipate continuing to provide telehealth music therapy services post-pandemic?”), and (7) perception of caregiver involvement (i.e., “Caregiver involvement is beneficial for a telehealth model”).

#### 2.6.4. Qualitative Analysis

The qualitative data (i.e., responses from open-ended questions) were interpreted using thematic analysis [39]. The process of coding the themes consisted of three of the authors becoming familiar with the data and determining common topics and followed the six-phase process for thematic analysis as outlined by Braun and Clark [39], which involves: becoming familiar with the data, generating initial codes, searching for themes, reviewing themes, defining the themes, and writing the report. Broader themes were then identified, examined, reconsidered, and altered until the final themes were determined. For the purposes of reliability and credibility, themes were also verified by the principal investigators and collaborators.

## 3. Results

### 3.1. Participants

A total of 573 music therapists responded to the survey. One respondent was excluded from the analysis as their participation was discontinued before completing the mandatory fields. Therefore, 572 responses were included in the analysis. The mean age (±standard deviation) of respondents was 41.3 ± 12.8, ranging from 23 to 85 years. The majority (82.7%) identified as female, followed by male (14%), non-binary/non-conforming (1.7%), prefer not to say (1.2%), and other (0.3%). Participants included individuals from 29 countries, with the top three highest percentages of music therapists currently living and practicing in the U.S. (67.5%), Canada (10.5%), and Australia (9.1%). Most respondents reported having a master’s degree as their highest level of education (53.7%), while a bachelor’s degree was the second most prevalent response (32.5%). It is important to note the terminology concerning education levels may vary across countries; for example, a bachelor’s degree category also includes participants that have an equivalency degree or/and a second bachelor’s degree. Regarding years of practicing, over half of the participants (52.2%) reported practicing as a music therapist for less than 10 years. Table 1 summarizes respondents’ demographic information. (See Appendix A: Survey Responses for a complete list alongside graphs and tables). 

### 3.2. Clinical Practice

The top four responses regarding therapists’ theoretical orientations were humanistic (27.9), eclectic and/or integrative (27.1%), neurologic music therapy (12%), and music psychotherapy (11.7%) representing 78.7% of participants’ responses. Primary work settings included private practice (43.7%), healthcare/medical facility (23.8%), academia (9.6%) including primary/secondary and tertiary/higher education combined, community music/arts center (5.8%), military or veteran medical center (3.0%), specialized clinic (2%), currently unemployed (1.6%), and other (10.5%).

Prior to COVID-19, the three primary clinical populations that respondents served were children with developmental delays (20.1%), persons with intellectual/multiple disabilities (18.3%), and mental health patients (11.7%). Other populations included older adults (10%), individuals with dementia (7.2%), adolescents (6.5%), clients/consumers in palliative/end of life care (6.2%), neurorehabilitation (5.8%), medical/oncology (4.4%), other (3.4%), and premature infants (1.3%). The responses regarding clinical populations being served since COVID-19 remained consistent with those prior to COVID-19, with only slight differences in percentages (see Appendix A).

With respect to the number of clinical hours participants worked each week prior to and during the COVID-19 pandemic, a Wilcoxon signed-rank test indicated that the number of clinical hours per week was significantly lower since the COVID-19 pandemic compared to before: *z* = −2.553, *p* = 0.011, *r* = −0.07. The results indicate that the transition to telehealth due to COVID-19 resulted in a reduced number of clinical hours, as shown in Figure 1.

### 3.3. Telehealth Provision

Participants used the following terms to describe remote music therapy delivery: TMT (45.8%); virtual music therapy (25.9%); online music therapy (20.1%); remote music therapy (4.4%); distance delivery of music therapy (0.3%); and other (3.5%). Only 12.8% of participants had provided TMT prior to the pandemic and about half of the music therapist respondents (50.2%) indicated that they had participated in education/training to develop skills for TMT, while 49.8% reported they did not. 

With respect to client goals, 42.1% of participants noted there were changes to goal areas when moving from in-person to TMT sessions. The most common clinical needs addressed with TMT were emotional expression (21.2%), speech and language/communication (16.6%), mood (13.6%), anxiety (12.6%), cognition (12.5%), isolation (11.2%), motor function (6.2%), spiritual support (1.9%), other (3%), and pain (1.1%). Music therapists indicated changes in their client’s goal areas during TMT sessions compared to in-person sessions. Goal changes were reported as definitely (7.9%), likely (21.2%), for some clients/consumers (42.1%), not at all (15%), and unsure (13.8%). The most prevalent interventions facilitated by music therapists during TMT sessions included: singing (27.2%), music listening (19.4%), songwriting (15.8%), movement to music (15.6%), improvisation (9.4%), mindfulness (7.3%), other (5.3%). The majority of participants indicated strong agreement regarding the successful use of live music (93%) and recorded music (74%) during in-person sessions (93%) as compared to TMT sessions, where about half indicated that they were able to successfully use live music (51%) and recorded music (57.5%). Regarding the perceived success of using live music, the Wilcoxon signed-rank test revealed that the success scores were significantly lower during TMT compared to in-person music therapy sessions, *z* = −14.439, *p* = 0.000, with a large effect size, *r* = −0.43. Similarly, the perceived success of using pre-recorded music was significantly lower during TMT compared to in-person sessions, *z* = −6.070, *p* < 0.001; however, there was a small effect size, *r* = −0.18. The results indicate that the transition to telehealth due to COVID-19 led to reduced perceived success rates in delivering both live and pre-recorded music as compared to in-person sessions.

Outcome measures used in TMT included observation (39.7%), client self-report (26.3%), family/caregiver report (24.1%), standardized assessment (7.4%), and assessments performed by other healthcare workers (2.5%). The ability to administer assessments over telehealth was reported as: agree/strongly agree (58.7%); neither agree nor disagree (27.1%); and disagree/strongly disagree (14.1%)

The most commonly reported platforms used by respondents for TMT included Zoom (59.1%), Facetime (8.3%), Teams (7.7%), Google Duo (6.3%), and Skype (4.8%). Music therapists ranked client privacy and confidentiality compliance (i.e., HIPAA) in their respective countries as compliant (75.9%), not compliant (2.4%), and not sure (21.7%). 

#### 3.3.1. Influence of Primary Theoretical Orientation and Work Setting

A chi-square analysis showed that there was a significant relationship between primary theoretical orientation and experience providing TMT prior to the pandemic, *X*^2^ (4, *n* = 572) = 32.12, *p* < 0.001, as well as perceptions of caregiver involvement, *X*^2^ (8, *n* = 572) = 43.78, *p* < 0.001. Post hoc comparisons using Bonferroni correction indicated that respondents who selected music psychotherapy as a primary theoretical orientation had significantly more experience in providing TMT prior to the pandemic compared to other theoretical approaches (*p.adj* = 0.00002) and the lowest percentage of agreement regarding the benefits of caregivers’ involvement for a telehealth model (*p.adj* = 0.000). Furthermore, the Pearson’s chi-square test by Monte Carlo Method revealed a significant association between primary theoretical orientation and global region of respondents, *p* = 0.006, 99% *CI* [0.004, 0.008], and their education level, *p* = 0.000, 99% *CI* [0.000, 0.001]. More specifically, the concentration of music therapists whose primary theoretical orientation was humanistic was significantly greater in North America (*p* = 0.002) compared to other global regions. Post hoc tests further showed that music psychotherapy orientation was most prevalent in the PhD cohort (*p.adj* = 0.00229), while humanistic orientation was most prevalent in the bachelor’s degree cohort (*p.adj* = 0.00081). However, there was no significant association between primary theoretical orientation and change in the client’s goal areas, *X*^2^ (8, *n* = 572) = 9.74, *p* = 0.284, perceptions of providing TMT services post-pandemic, *X*^2^ (8, *n* = 572) = 11.51, *p* = 0.174, and years practicing as a music therapist, *X*^2^ (8, *n* = 572) = 13.69, *p* = 0.090.

An additional chi-square test of independence revealed that there was a significant relationship between primary work setting and education level, *X*^2^ (9, *n* = 571) = 67.11, *p* < 0.001, years practicing as a music therapist, *X*^2^ (6, *n* = 572) = 18.45, *p* = 0.005, experience with TMT prior to the pandemic, *X*^2^ (3, *n* = 572) = 10.67, *p* = 0.014, perceptions of caregiver involvement, *X*^2^ (6, *n* = 572) = 32.29, *p* < 0.001, and perceptions of providing TMT services post-pandemic, *X*^2^ (6, *n* = 572) = 21.54, *p* = 0.001. No significant association was found between primary work setting and change in client’s goal areas due to the transition to telehealth, *X*^2^ (6, *n* = 572) = 12.49, *p* = 0.052, and global region of respondents, *p* = 0.182, 99% *CI* [0.172, 0.192]. Post hoc tests further indicated that the largest number of respondents with a doctorate (*p.adj* = 0.00000), as well as the largest number of respondents with more than 20 years of work experience (*p.adj* = 0.00032), was found among those working in academia (*p.adj* = 0.00000). However, no significant association was found between primary work setting and experience with TMT prior to the pandemic following the Bonferroni correction. 

Regarding perceptions of caregiver involvement (i.e., “Caregiver involvement is beneficial for a telehealth model”), those working in private practice had the lowest percentage of disagreement with this statement (*p.adj* = 0.00298), whereas respondents working in a healthcare/medical facility most commonly selected “neither agree nor disagree” (*p.adj* = 0.00194). Finally, when asked: “Do you anticipate continuing to provide TMT services post-pandemic?”, music therapists primarily working in private practice selected “Likely” or “Yes, definitely” significantly more than other respondents (*p.adj* = 0.00339), while those working in academia most commonly selected “Not at all” or “For some clients/consumers” (*p.adj* = 0.00385). From these results, we posit that music therapists working in private practice are most inclined to continue to provide TMT services post-pandemic compared to those in other work settings, especially academia.

#### 3.3.2. Telehealth Perceptions

Regarding perceptions of TMT having more benefits than drawbacks, music therapists differed in their responses, as indicated in Figure 2, with 15.4% strongly agreeing, 26.6% agreeing, 36.7% neither agreeing or disagreeing, 16.6% disagreeing, and 4.7% strongly disagreeing. This aligns with the participants’ intent to continue TMT practices post-pandemic, as shown in Figure 3, wherein 39.3% indicated that they will continue, 19.6% reported that they will continue using TMT with some clients/consumers, 18.2% stated they will likely continue, 12.2% were unsure, and 10.7% indicated no intent to continue.

A key finding regarded caregiver involvement and its benefits for TMT, as demonstrated in Figure 4. The majority of music therapists (41.1%) strongly agreed, 33.2% agreed, 22.4% neither agreed nor disagreed, and small percentages disagreed (2.3%), and strongly disagreed (1%) that involving caregivers in TMT sessions was beneficial.

### 3.4. Correlation Analysis

Table 2 shows the correlations between the selected variables. There were moderate-to-strong positive correlations between future telehealth provision and telehealth perceptions, *r_s_* (570) = 0.428, *p* < 0.001, future telehealth provision and ability to administer assessments, *r_s_* (570) = 0.399, *p* < 0.001, as well as telehealth perceptions and ability to administer assessments, *r_s_* (570) = 0.332, *p* < 0.001. In other words, music therapists who perceived telehealth music therapy to have more benefits than drawbacks also reported being more proficient at administering assessments over telehealth and had a higher perceived likelihood of using telehealth in the future as compared to music therapists with a less positive perception of TMT. Furthermore, weak but significant correlations were found between years practicing as a music therapist and education level, *r_s_* (569) = 0.211, *p* < 0.001, years practicing and telehealth perceptions, *r_s_* (570) = 0.138, *p* < 0.001, years practicing and future telehealth provision, *r_s_* (570) = 0.109, *p* = 0.009, as well as education level and future telehealth, *r_s_* (569) = 0.168, *p* < 0.001, and ability to administer assessments, *r_s_* (569) = 0.089, *p* = 0.034. That is, music therapists with more years of work experience had more positive perceptions of telehealth, a higher perceived likelihood of using telehealth in the future, and higher education levels. Higher education levels were also associated with a greater perceived ability to administer assessments.

Additionally, changes in the client’s goal areas (TMT vs. in-person) were negatively correlated with years practicing as a music therapist, *r_s_* (570) = −0.101, *p* = 0.016, telehealth perceptions, *r_s_* (570) = −0.122, *p* = 0.003, ability to administer assessments, *r_s_* (570) = −0.188, *p* < 0.001, and future telehealth, *r_s_* (570) = −0.148, *p* < 0.001. This indicates that music therapists who implemented more changes in the client’s goal areas when transitioning from in-person sessions to telehealth had fewer years of experience, a less positive perception of telehealth, a lower perceived ability to administer assessments via telehealth, and a lower perceived likelihood of using telehealth in the future. Lastly, the perception of caregiver involvement had significant but weak positive correlations with telehealth perceptions, *r_s_* (570) = 0.095, *p* = 0.023, and changes in client’s goal areas, *r_s_* (570) = 0.091, *p* = 0.030. This indicates that clinicians who perceived caregiver involvement as being beneficial for TMT also had more positive perceptions of telehealth provision, made more changes in the client’s goal areas when transitioning from in-person sessions to telehealth, and had more positive perceptions of telehealth provision.

### 3.5. Qualitative Analysis

Qualitative analysis rendered three overall themes with respect to training, benefits, and challenges for clients/consumers and for music therapists. 

#### 3.5.1. Training Taken by Music Therapists to Prepare for TMT Delivery

Almost half (49.8%) of the respondents reported taking additional training, workshops, and/or courses to develop their skills or increase their proficiency of TMT service provision. Of those who engaged in additional training, 96% provided further specifications of the additional training they attended. Additional training took a variety of formats, including training offered by music therapy organizations (*n =* 50/18.3%), national music therapy associations (*n =* 32/11.7%), conference attendance and presentations (*n =* 20/7.3%), training and webinars offered by employer or organization (*n =* 16/5.8%), training offered by non-music therapy organizations (*n* = 11/4%) and consulting the literature (*n =* 5/1.8%).

#### 3.5.2. Benefits and Challenges for Clients/Consumers Engaging in a TMT Session

For clients/consumers, increased access to music therapy was the main benefit reported (*n =* 161/31.6%), as travel was not a requirement (*n* = 33/6.4%). The continuation of music therapy during a time of uncertainty (i.e., COVID-19) was highlighted (*n* = 111/21.8%) as an important benefit, as well as the comfort of remaining in their own home (*n* = 64/12.5%) and reducing social isolation (*n* = 56/11%). Convenience (5%), flexibility (5%), and ease of scheduling (4.5%) were also reported.

Access to technology and internet connection were the most reported challenges for clients/consumers *(n* = 127/24.2%), with connection issues being the most significant problem (*n* = 89/16.9%). The unavailability of instruments was also a challenge (*n* = 68/12.9%). Clients/consumers’ disinterest and inability to focus on the screen for the full duration of the session was also reported (*n* = 66/12.5%). This appeared to be linked to lack of motivation to engage, screen fatigue, and clients/consumers who experience cognitive (e.g., memory) or behavioral (e.g., hyperactivity) issues. Regarding technological challenges, several factors were identified, including latency issues (8.7%), poor sound quality (5.4%), inability to make music synchronously (4%), inability or limited ability to use technology (2.9%), requiring assistance with set-up (1.7%), and device quality/inappropriate devices (*n* = 8/1.5%). Privacy issues and environmental distractions were also reported as challenging (*n* = 23/4.3%). 

#### 3.5.3. Benefits and Challenges for Music Therapists Engaging in TMT

The main benefits reported for music therapists were reduced travel time (*n* = 117/21.3%), continuity of service (*n* = 82/16.3%), the ability to work from home (*n* = 55/10.9%), flexibility in scheduling (*n =* 49/9.7%) and the exploration of new technologies and development of technical skills (*n =* 17/3%). The ability to provide basic training and develop relationships with parents/caregivers was also mentioned as being beneficial (*n =* 15). The main challenges reported by music therapists included: poor sound quality (*n* = 77/14.9%); latency issues resulting in the inability to improvise synchronously (*n* = 27/5.2%); difficulty in maintaining the clients/consumers engagement (*n* = 65/12.6%); fatigue, exhaustion and burnout (*n* = 41/7.9%); lack of access to musical instruments (*n* = 39/7.5%); and lack of response or difficulty in identifying non-verbal cues (*n* =19/3.6%).

## 4. Discussion

Considering the immediacy with which therapists pivoted to TMT due to the COVID-19 pandemic, it was helpful to glean the experiences of a global community regarding this shift in practice. Open-ended survey responses provided insights into the adaptations required to ensure the continuity of treatment when moving from an in-person to a virtual clinical environment and deepened the understanding of the benefits and challenges of TMT service provision for both therapists and clients/consumers. Many of the survey responses pertaining to primary areas of overarching concern and challenges with TMT were aligned with the literature; for example, the technical issues noted by study participants were also found by Agres et al. [23] and Baker and Tamplin [24]. These themes included navigating technological issues such as internet connectivity, the user-friendliness of platforms, audio/visual quality/latency, screen/image size, ensuring privacy/confidentiality, and the use of appropriate settings that support a full and qualitative musical experience. Additional challenges that were reported were screen fatigue and burn-out, need for caregiver support, setting boundaries with clients/consumers and caregivers, differentiation in contact (e.g., visual, physical), and the impact under clinical observation. Further reported challenges included the ambiguity of being accurately seen, heard, and perceived; maintaining empathy, and the physical and emotional positioning of therapist and client; the ability to address needs and specific goals; and limited interpersonal and non-verbal communication. 

Again, aligned with the literature, the primary reported benefits of TMT for both music therapists and clients/consumers included the ability to continue music therapy practices, continuity of care, increased creative thinking, flexibility of scheduling, sustained momentum of treatment, and caregiver investment in and advocacy of clients/consumers’ care. The extended reach regarding client base, treatment adaptation and acquisition of new skills that may be useful in ongoing practice, as well as the ability to leverage technology for use in innovative ways and integrate various music technologies and communication platforms, and the decreased health and safety risks, were also reported. An additional insight provided by survey respondents regarding modifications to TMT was that, despite the many changes in its appropriate implementation, most interventions were adaptable to the degree that they could be successfully presented and received, which preserved therapeutic success in the TMT space. These results echo the earlier work of Kantorová et al. [26], who noted that the benefits of TMT were increased access to clients/consumers and reaching clients/consumers that may have barriers to in-person treatment. 

Simultaneous active music-making, singing, and improvisation were cited by respondents as the most difficult to adapt; however, it was reported that overcoming these challenges was beneficial for the clients/consumers and therapists as it strengthened trust and rapport. Opportunities for continued engagement in physically distant yet socially connected spaces were instrumental to clinical gains, especially in the psychosocial realm. Lastly, it would be an oversight not to address how the lack of access to technological equipment, instruments and hands-on experiential treatment approaches impacted TMT treatment. This includes the high costs of technological devices and internet access in some regions, which made the basic set-up of TMT less affordable for music therapists in lower-income countries. On one hand, some clients/consumers did not have access to the variety of instrumentation that they would in an in-person music therapy setting. On the other hand, some music therapists reported that they innovated new instrument loan procedures and processes for instrument creation with household items and other found objects that they found helpful, and aimed to continue using in their practices moving forward. 

Taking strength-based approaches to music therapy treatment, whether in-person or via TMT, is essential to treatment success. In addition to the clinical space, respondents also reported how shifts to virtual environments impacted education, training, and supervision. Educators and supervisors reported spending ample time checking in with students and supervisees individually and in smaller groups, which corresponds to Gooding and Rushing’s findings [34]. Similar to the findings of Kern and Tague [33], students and interns reported that there were certain skill sets that were more difficult to develop in TMT, such as in the moment musicking and developing effective clinical observation skills without seeing ‘the whole picture’. Conversely, some respondents reported that as long as the technology functions as intended, online supervision is effective and convenient and that they can see a role for hybrid practices in the future of education and training curriculums. 

### 4.1. Limitations

This study has some limitations. The survey was published in English only, which limited the number of international respondents to those who were fluent in English. Respondents were required to have only facilitated 10 TMT sessions, so insights may have shifted from continued practice with TMT. Almost half of the respondents were employed via private practice; therefore, they may have not had access to resources or support to assist themselves and their clients/consumers in the transition from in-person music therapy to TMT settings. Further, it is possible that music therapists who participated in this survey might be more interested in building or developing TMT skills. 

### 4.2. Recommendations

Based on the results of this survey, it is recommended that music therapists utilizing TMT within their practice continue to assess the suitability of this delivery for each individual client, and that a hybrid programming combining in-person and TMT may be optimal to support client access and engagement. Music therapists should implement creative thinking, flexibility in facilitation, and an innovative adaptation to practices, as well as engaging in continued learning, increasing their technological literacy, and reading the current on literature on this topic to ensure adherence to ethical practice. Further, an ongoing awareness of the changing guidelines to ensure compliance with local laws and ethical standards is needed to maintain client privacy and confidentiality over time.

Further recommendations for clinical telehealth training include training in cultural competencies and cultural humility, as music therapists are increasingly being called to provide services to clients/consumers from diverse cultural backgrounds. To increase inclusivity, TMT training is needed to allow for music therapists to effectively deliver services on a variety of technological platforms. For example, increasing access to TMT in some regions may require that music therapists know how to provide effective TMT delivery on mobile devices. Additionally, translating TMT training materials into other languages may enable non-English speaking music therapists to obtain broader access to TMT as an option for delivering their services. Widening music therapists’ telehealth competencies and skill sets to include video, social media, innovative technologies, digital applications, and platforms for asynchronous connection to optimize their service delivery will be helpful. Further research including client perspectives of telehealth music therapy would better inform future practice in this area.

A prevailing benefit of TMT reported in prior studies is access to music therapy services by populations that, either through geography, safety concerns, or mobility issues, would otherwise not have access to in-person music therapy. TMT is a promising intervention to increase access to populations in which these factors are barriers to participation in in-person music therapy. Therefore, it is recommended to continue providing opportunities for clients/consumers to connect to treatment either remotely or through hybrid forms of treatment. 

Kantorová et al. [26] inferred that the challenges in transferring between in-person services and virtual sessions suggested that “music as a therapeutic agent doesn’t work in the same way” and that “music therapy as a relation therapy may lose important aspects based on physical contact when transferred to the virtual space” (p. 11). However more recently, Cephas et al. [31] reported specific digital applications that may enhance musical interactions and facilitate therapeutic outcomes. Future research should identify service users who demonstrate increased engagement through digital applications and whose clinical needs may be best met through TMT. Additionally, the development and use of screening tools for music therapists to assess telehealth readiness, such as that introduced by Richard Williams et al. [40] for clients/consumers with autism, provide systematic approaches that may address these gaps in the delivery of TMT.

## 5. Conclusions

In 2021, the Certification Board for Music Therapy suggested that “telepractice is here to stay” [41]. Reinforcing this, a majority of respondents to our study reported they will continue using TMT. What was initially a necessary adaptation out of concern for public health and safety has become a new service delivery model that promises increased access to music therapy for those with barriers to in-person treatment and for those that may uniquely benefit from the remote service delivery format. Music therapists who approach telehealth as a distinctly intimate and resource-rich service delivery format rather than a secondary alternative to in-person services may be more successful in utilizing the resources inherent in the digital space they share with their clients/consumers, leading to optimized therapeutic benefits. 

Existing TMT research insufficiently addresses cultural factors, access, and affordability. Further recommendations for research should also explore the development of skills and competencies in the aforementioned areas. Research into TMT outreach and advocacy for often neglected and marginalized groups also needs to be prioritized as a health concern in the interests of embracing diversity, equity, inclusion, and accessibility across the globe. Last but not least, continuous engagement between music therapists, researchers, policy makers, and administrators is needed to ensure that quality health care, including TMT, is cost-effective and affordable for all.

## Figures and Tables

**Figure 1 ijerph-20-05580-f001:**
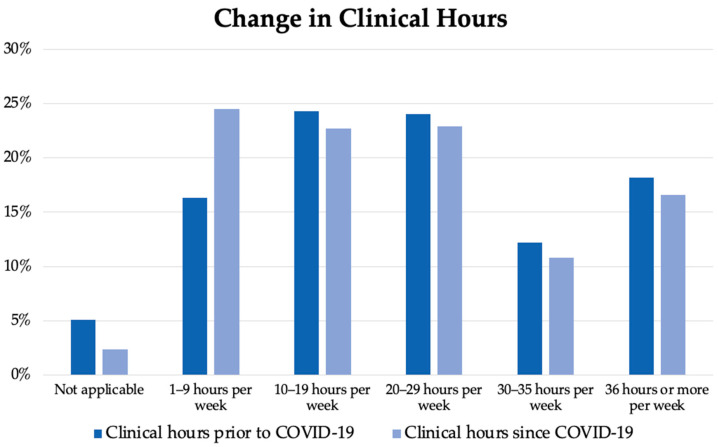
Change in clinical hours prior to and since COVID-19.

**Figure 2 ijerph-20-05580-f002:**
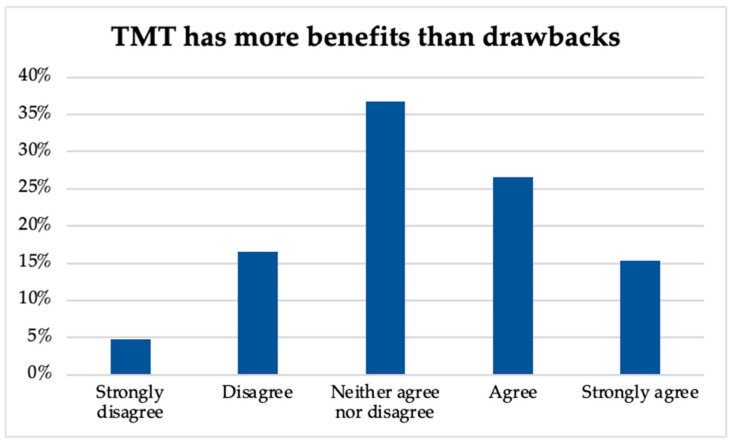
TMT benefits vs. drawbacks.

**Figure 3 ijerph-20-05580-f003:**
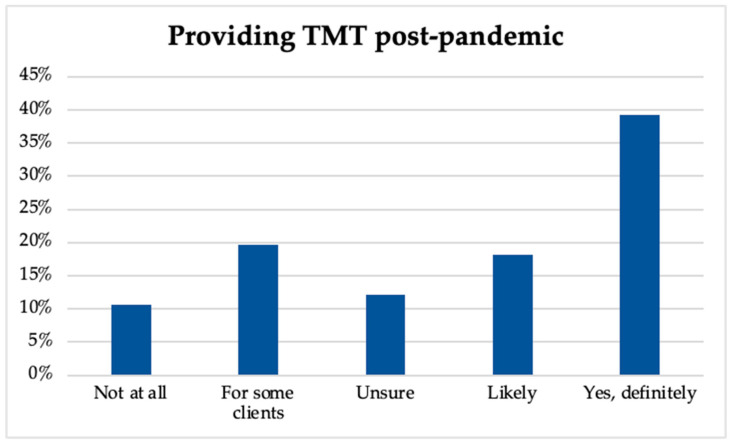
Providing TMT post-pandemic.

**Figure 4 ijerph-20-05580-f004:**
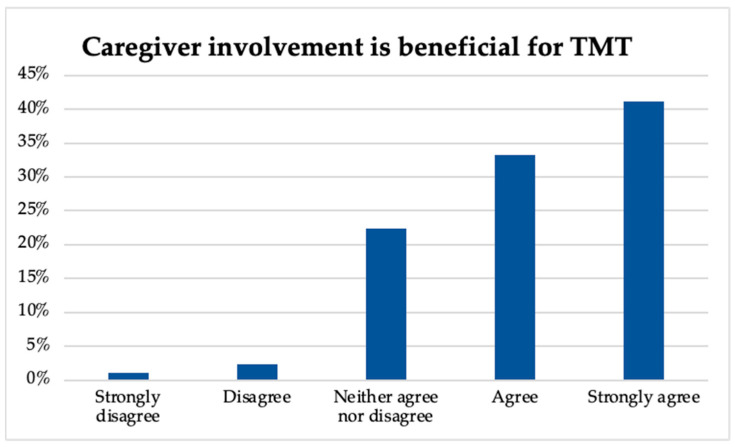
Perceptions of caregiver involvement during TMT.

**Table 1 ijerph-20-05580-t001:** Respondent demographics (*n* = 572).

Variables	Mean (SD)/*n* (%)
**Age**	41.3 (±12.8)
**Gender**	
Female	473 (82.7%)
Male	80 (14%)
Non-binary/non-conforming	10 (1.7%)
Prefer not to say	7 (1.2%)
Other	2 (0.3%)
**Country**	
United States	386 (67.5%)
Canada	60 (10.5%)
Australia	52 (9.1%)
Other	74 (12.9%)
**Country per region**	
North America	446 (78%)
Oceania	57 (10%)
Europa	33 (5.8%)
Asia	26 (4.5%)
South America	10 (1.7%)
**Education level**	
Bachelor’s degree	186 (32.5%)
Master’s degree	307 (53.7%)
Doctorate	54 (9.4%)
Graduate Certificate	12 (2.1%)
Other	12 (2.1%)
**Years practicing as a music therapist**	
Less than 5 years	146 (25.5%)
5–10 years	152 (26.6%)
10–15 years	93 (16.3%)
15–20 years	52 (9.1%)
More than 20 years	129 (22.6%)

**Table 2 ijerph-20-05580-t002:** Correlations analysis table.

	1	2	3	4	5	6	7
(1) Years practicing	1						
(2) Telehealth perceptions	0.138 ** <0.001	1					
(3) Education level	0.211 ** <0.001	0.077 0.066	1				
(4) Change in client’s goal areas (TMT vs. in-person)	−0.101 * 0.016	−0.122 ** 0.003	−0.069 0.100	1			
(5) Ability to administer assessments	−0.017 0.679	0.332 ** <0.001	0.089 * 0.034	−0.188 ** <0.001	1		
(6) Future telehealth	0.109 ** 0.009	0.428 ** <0.001	0.168 ** <0.001	−0.148 ** <0.001	0.399 ** <0.001	1	
(7) Perception of caregiver involvement	−0.021 0.615	0.095 * 0.023	−0.068 0.104	0.091 * 0.030	−0.027 0.522	−0.037 0.381	1

* *p* < 0.05, ** *p* < 0.01 (2-tailed).

## Data Availability

The data presented in this study are available on request from the corresponding author. The data are not publicly available due to ensure confidentiality.

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
