# Peer review of "International Music Therapists’ Perceptions and Experiences in Telehealth Music Therapy Provision"

_ijerph, 2023, doi:10.3390/ijerph20085580_

Round 1

Reviewer 1 Report

Thanks for the opportunity to review this manuscript reporting the results of an international survey on telehealth music therapy practice. The paper is well-written and addresses the stated research questions adequately. There are a few specific areas that are unclear or need minor revision in my opinion. I have outlined these below:

Abstract

“The results showed that the number of clinical hours declined due to the pandemic.” This sentence is a little vague. Do you mean overall music therapy hours decreased (including both face to face and TMT) or face to face therapy only decreased?

Suggest revising this sentence as it seems incomplete: “Rising to the challenges posed by the pandemic, there was no clear agreement on whether TMT has more benefits than drawbacks…” perhaps something like: “Although many music therapists rose to the challenges posed by the pandemic by incorporating TMT delivery modes, there was no clear agreement on whether TMT has more benefits than drawbacks…”

Introduction

There seems to be a word missing on p2: “this article will refer to the remote delivery of healthcare services as telehealth and the remote delivery of music therapy as telehealth music therapy (TMT).”

Literature

There seems to be some missing words in this sentence on p3 about the Baker and Tamplin survey.

“Benefits were across categories of safety, financial, client, service delivery, and time management. Reported challenges included technical, session design, facilitation, and the therapeutic relationship.”

Can you be more specific about what the benefits re clients were? Was this to do with engagement for example? And what were the benefits re service delivery? Similarly with challenges, some of these require further explanation, eg. maybe “technical difficulties”? And what was it about session design, facilitation, and the therapeutic relationship that were reported as challenging?

This sentence on p3 needs revising for grammar: “Wilhelm and Wilhelm (2022) surveyed music therapists providing TMT to older adults, which yielded similar results...”

Suggested changes: “Wilhelm and Wilhelm (2022) conducted a survey of music therapists providing TMT to older adults, which yielded similar results…”

Another sentence on p3 needs revising regarding the Kantorová study. It’s also unclear what the proximity challenge specifically relates to: “Challenges included synchronization of musical interactions, sound quality, and the proximity and lack of ability to see and touch the instruments.”

Suggested changes: “Challenges included inability to synchronize musical interactions, poor sound quality, and the proximity to the therapist/instruments? and lack of ability to see and touch the instruments.

The following sentence on p3 refers to “articles” plural, but only cites one article by Knott & Block (2020).

Method

I’m not sure that the study design is described accurately. The authors state that this is a phenomenological mixed methods study, but I see no evidence of phenomenology or mixed methods processes. There is no description of a mixed methods design as concurrent/sequential or any attempt to integrate the 2 types of data, which is the main feature of mixed methods research. The method of qualitative analysis is later described as thematic analysis, rather than a phenomenological method of analysing the open-ended survey responses. Perhaps it is more accurate to describe as a survey design study with a mixture of quantitative and qualitative data.

In the section on Chi-Square analysis, I’m not sure what you mean by: “When more than 20% of the expected counts were less than five…”

Results

On p8, in the last line of the Telehealth Provision section, did you mean counties or countries?

The sentences on post hoc analyses at the top of p9 are worded in a strange way:

“the largest number of music therapists who hold a doctoral degree had selected music psychotherapy as a primary theoretical orientation whereas the largest number of music therapists with a bachelor’s degree was among the respondents practicing under a humanistic orientation.”

Could you just say that psychotherapeutic orientation was most prevalent in the PhD cohort and humanistic orientation was most prevalent in the bachelor’s degree cohort?

Discussion

P14. “instrument check-out procedures” could be better expressed as “instrument loan procedures”

The first line of the recommendations section is a little circular. “Recommendations for ongoing integration of TMT into practice includes the continuation of TMT…”. Consider revising.

Do you have a reference for this statement? “Populations in Asia have more access to and are more familiar with mobile phones and tablets rather than laptops or desktops.” If not, it sounds like a bit of a cultural assumption.

In the following sentence, ‘therapists’ needs a possessive apostrophe, and please define what you mean by mobile health and telebehavioural health: “Widening music therapists’ telehealth competencies and skill sets to leverage use of video, social media, mobile health, telebehavioral health, and asynchronous service delivery will be helpful.”

I’m not clear why there is a paragraph in the discussion describing the Ahessy study. Perhaps this is better placed in the literature review section. If the point you are making is that your survey didn’t capture client perspectives and that this would be good to do in future studies, then say this overtly. You can then reference Ahessy and Baker & Tamplin as examples of previous survey studies that have collected data on the participant perspective of TMT.

In the following paragraph on ‘access’, geographical location is not the only barrier to participation. In the preceding sentence you also mention safety concerns and mobility issues. Please ensure all of these are covered in the sentence on barriers to participation also.

Conclusion

As stated earlier, you haven’t defined what is meant by ‘mobile health’.

“Further recommendations for research should also discuss the development of skills and competencies in the aforementioned areas.” Do you mean: “Future research should also explore the development of skills and competencies in the aforementioned areas.”?

In the sentence talking about advocating for the needs of marginalised groups, promoting diversity doesn’t quite work. I think you mean embracing diversity by promoting equity, accessibility, and inclusion. Or something along these lines. Please revise as needed.

Author Response

Abstract

“The results showed that the number of clinical hours declined due to the pandemic.” This sentence is a little vague. Do you mean overall music therapy hours decreased (including both face to face and TMT) or face to face therapy only decreased?

Yes, we were referring to the overall hours. Sentence revised to: The results showed that the overall number of clinical hours (TMT and in-person hours combined) declined due to the pandemic.

Suggest revising this sentence as it seems incomplete: “Rising to the challenges posed by the pandemic, there was no clear agreement on whether TMT has more benefits than drawbacks…” perhaps something like: “Although many music therapists rose to the challenges posed by the pandemic by incorporating TMT delivery modes, there was no clear agreement on whether TMT has more benefits than drawbacks…”

 REVISED as per your Suggestion. Thanks.

Introduction

There seems to be a word missing on p2: “this article will refer to the remote delivery of healthcare services as telehealth and the remote delivery of music therapy as telehealth music therapy (TMT).”

Added remote

Literature

There seems to be some missing words in this sentence on p3 about the Baker and Tamplin survey.

“Benefits were across categories of safety, financial, client, service delivery, and time management. Reported challenges included technical, session design, facilitation, and the therapeutic relationship.”

Can you be more specific about what the benefits re clients were? Was this to do with engagement for example? And what were the benefits re service delivery? Similarly with challenges, some of these require further explanation, eg. maybe “technical difficulties”? And what was it about session design, facilitation, and the therapeutic relationship that were reported as challenging?

More specific information was added.

This sentence on p3 needs revising for grammar: “Wilhelm and Wilhelm (2022) surveyed music therapists providing TMT to older adults, which yielded similar results...”

Suggested changes: “Wilhelm and Wilhelm (2022) conducted a survey of music therapists providing TMT to older adults, which yielded similar results…”

 Revised.

Another sentence on p3 needs revising regarding the Kantorová study. It’s also unclear what the proximity challenge specifically relates to: “Challenges included synchronization of musical interactions, sound quality, and the proximity and lack of ability to see and touch the instruments.”

Suggested changes: “Challenges included inability to synchronize musical interactions, poor sound quality, and the proximity to the therapist/instruments? and lack of ability to see and touch the instruments.

 Revised as per suggestion.

The following sentence on p3 refers to “articles” plural, but only cites one article by Knott & Block (2020).

 Revised as per suggestion.

Method

I’m not sure that the study design is described accurately. The authors state that this is a phenomenological mixed methods study, but I see no evidence of phenomenology or mixed methods processes. There is no description of a mixed methods design as concurrent/sequential or any attempt to integrate the 2 types of data, which is the main feature of mixed methods research. The method of qualitative analysis is later described as thematic analysis, rather than a phenomenological method of analysing the open-ended survey responses. Perhaps it is more accurate to describe as a survey design study with a mixture of quantitative and qualitative data.

Agreed revised to descriptive survey design.

In the section on Chi-Square analysis, I’m not sure what you mean by: “When more than 20% of the expected counts were less than five…”

The conventional rule of thumb is that all of the expected numbers (data) in any given cell should be greater than 5 in order to use the chi-square analysis; otherwise, an alternative test should be used, such as an exact test of goodness-of-fit or a Fisher's exact test of independence. If less than 20% of the expected counts are less than five, it is still acceptable to use the chi-square. Therefore, only when more than 20% of the expected counts were less than five, we applied Fisher’s Monte Carlo Method, as suggested in the literature (Mehta & Patel, 2011).

Results

On p8, in the last line of the Telehealth Provision section, did you mean counties or countries?

 Thank you for catching this. It is countries and is revised now.

The sentences on post hoc analyses at the top of p9 are worded in a strange way:

“the largest number of music therapists who hold a doctoral degree had selected music psychotherapy as a primary theoretical orientation whereas the largest number of music therapists with a bachelor’s degree was among the respondents practicing under a humanistic orientation.”

Could you just say that psychotherapeutic orientation was most prevalent in the PhD cohort and humanistic orientation was most prevalent in the bachelor’s degree cohort?

Thank you. We agree, our attempt was to use the same language as in the previous sections, but we revised as per suggestion: In our sample, music psychotherapy orientation was most prevalent in the PhD cohort (p.adj = .00229), while humanistic orientation was most prevalent in the bachelor’s degree cohort (p.adj = .00081).

Discussion

P14. “instrument check-out procedures” could be better expressed as “instrument loan procedures”

 Thank you that is better wording. We have revised.

The first line of the recommendations section is a little circular. “Recommendations for ongoing integration of TMT into practice includes the continuation of TMT…”. Consider revising. 

 Revised “Based on the results of this survey it is recommended that music therapists utilizing TMT within their practice continue to assess the suitability of this delivery for each individual client, and that hybrid programming combining in-person and TMT may be optimal to support client access and engagement”

Do you have a reference for this statement? “Populations in Asia have more access to and are more familiar with mobile phones and tablets rather than laptops or desktops.” If not, it sounds like a bit of a cultural assumption.

 Sentence removed.

In the following sentence, ‘therapists’ needs a possessive apostrophe, and please define what you mean by mobile health and telebehavioural health: “Widening music therapists’ telehealth competencies and skill sets to leverage use of video, social media, mobile health, telebehavioral health, and asynchronous service delivery will be helpful.”

 Sentence Revised.

I’m not clear why there is a paragraph in the discussion describing the Ahessy study. Perhaps this is better placed in the literature review section. If the point you are making is that your survey didn’t capture client perspectives and that this would be good to do in future studies, then say this overtly. You can then reference Ahessy and Baker & Tamplin as examples of previous survey studies that have collected data on the participant perspective of TMT.

Removed.

In the following paragraph on ‘access’, geographical location is not the only barrier to participation. In the preceding sentence you also mention safety concerns and mobility issues. Please ensure all of these are covered in the sentence on barriers to participation also.

 Sentence revised.

Conclusion

As stated earlier, you haven’t defined what is meant by ‘mobile health’.

 Term Removed.

“Further recommendations for research should also discuss the development of skills and competencies in the aforementioned areas.” Do you mean: “Future research should also explore the development of skills and competencies in the aforementioned areas.”?

 Explore is the word. Revised. Thanks.

In the sentence talking about advocating for the needs of marginalised groups, promoting diversity doesn’t quite work. I think you mean embracing diversity by promoting equity, accessibility, and inclusion. Or something along these lines. Please revise as needed.

Revised as per your suggestion. Thanks.

Reviewer 2 Report

This article is difficult to assess. The topic is timely given the context of post-Covid concerns and the need and viability of telehealth issues even after the end of the epidemic. The paper is well written and the methodology seems to be sound, but after reading the question raises as to the real take home message. Are the findings really substantial and do they provide new insights beyond some predictable and general intuitions? Are they strong enough to function as material to be published in a high standards academic journal? I doubt. 

General comments

-      The language use is OK and the style of writing is fluent. 

-      The methodology is sound and the number of participants (n = 572) is quite large, which gives the paper some generalizing power. 

-      The list of references is OK, though some more foundational papers could be inserted. 

-      What is lacking somewhat is a broader theoretical background and positioning of the concepts of telemedicine, telerehabilitation, teleintervention, and telehealth. This should be elaborated more in depth. 

-      The discussion provides some interesting findings.

Detailed comments

-      The abstract is OK and is well-written.

-      Page 2, 3rd paragraph: here, the concept of telehealth and the related concepts should be elaborated more in detail.

-      Page 4, penultimate par.: What is meant exactly with a scoping review? Explain somewhat more in detail. 

-      Page 4, 2nd par. What is meant exactly with phenomenological mixed methods study. Explain shortly in intuitive terms.

-      Page 5, 2nd par. What does the abbreviation PIs stand for?

-      Page 6: more should be said about the thematic analysis. Explain more in detail how it was done and what were the major findings.

-      Page 6, table 1. The number of participants is a sample and not a population. What to use: N = 872 or n = 572?

Author Response

This article is difficult to assess. The topic is timely given the context of post-Covid concerns and the need and viability of telehealth issues even after the end of the epidemic. The paper is well written and the methodology seems to be sound, but after reading the question raises as to the real take home message. Are the findings really substantial and do they provide new insights beyond some predictable and general intuitions? Are they strong enough to function as material to be published in a high standards academic journal? I doubt. 

We appreciate your review. Given this study is specific to music therapy and is a large representation of the profession, we believe the results add considerable value to the small body of emerging research.

General comments

-      The language use is OK and the style of writing is fluent. 

-      The methodology is sound and the number of participants (n = 572) is quite large, which gives the paper some generalizing power. 

-      The list of references is OK, though some more foundational papers could be inserted. 

We have done a large review and feel the articles included are the most relevant. We are also cognizant of the space and word limits in increasing our literature review.

-      What is lacking somewhat is a broader theoretical background and positioning of the concepts of telemedicine, telerehabilitation, teleintervention, and telehealth. This should be elaborated more in depth. 

-      The discussion provides some interesting findings.

Detailed comments

-      The abstract is OK and is well-written.

-      Page 2, 3rd paragraph: here, the concept of telehealth and the related concepts should be elaborated more in detail.

We feel enough detail is provided as it is beyond the scope of the paper to do an extensive history of the praxis.

-      Page 4, penultimate par.: What is meant exactly with a scoping review? Explain somewhat more in detail. 

We feel that readers would know what a scoping review is and that it is beyond the scope of the paper to provide a definition of this here.

-      Page 4, 2nd par. What is meant exactly with phenomenological mixed methods study. Explain shortly in intuitive terms.

We have revised the method to descriptive survey

-      Page 5, 2nd par. What does the abbreviation PIs stand for?

Principle investigators. Revised.

-      Page 6: more should be said about the thematic analysis. Explain more in detail how it was done and what were the major findings.

Added information on the Braun and Clark 6 phase process. We feel the findings are written about in the results section that follows.

-      Page 6, table 1. The number of participants is a sample and not a population. What to use: N = 872 or n = 572?

Thank you. Corrected to n = 572.

Round 2

Reviewer 2 Report

Most of the previous comments have been addressed appropriately. I therefore suggest to accept the paper for publication.